# Kub3 Deficiency Causes Aberrant Late Embryonic Lung Development in Mice by the FGF Signaling Pathway

**DOI:** 10.3390/ijms23116014

**Published:** 2022-05-27

**Authors:** Guangying Yang, Shan Lu, Jia Jiang, Jun Weng, Xiaomei Zeng

**Affiliations:** Key Laboratory of Molecular Biophysics of Chinese Ministry of Education, Center for Human Genome Research, College of Life Science and Technology, Huazhong University of Science and Technology, Wuhan 430074, China; d201780464@hust.edu.cn (G.Y.); m201971907@hust.edu.cn (S.L.); m202071986@hust.edu.cn (J.J.)

**Keywords:** Kub3, embryonic lung development, pulmonary atelectasis, cell differentiation, FGF signaling pathway

## Abstract

As a Ku70-binding protein of the KUB family, Kub3 has previously been reported to play a role in DNA double-strand break repair in human glioblastoma cells in glioblastoma patients. However, the physiological roles of Kub3 in normal mammalian cells remain unknown. In the present study, we generated *Kub3* gene knockout mice and revealed that knockout (KO) mice died as embryos after E18.5 or as newborns immediately after birth. Compared with the lungs of wild-type (WT) mice, *Kub3* KO lungs displayed abnormal lung morphogenesis and pulmonary atelectasis at E18.5. No difference in cell proliferation or cell apoptosis was detected between KO lungs and WT lungs. However, the differentiation of alveolar epithelial cells and the maturation of type II epithelial cells were impaired in KO lungs at E18.5. Further characterization displayed that Kub3 deficiency caused an abnormal FGF signaling pathway at E18.5. Taking all the data together, we revealed that Kub3 deletion leads to abnormal late lung development in mice, resulting from the aberrant differentiation of alveolar epithelial cells and the immaturation of type II epithelial cells due to the disturbed FGF signaling pathway. Therefore, this study has uncovered an essential role of Kub3 in the prenatal lung development of mice which advances our knowledge of regulatory factors in embryonic lung development and provides new concepts for exploring the mechanisms of disease related to perinatal lung development.

## 1. Introduction

The lung is an important organ for aerobic respiration in vertebrates. During the embryonic development stage, the lung goes through a series of structural and functional development and maturation processes to prepare for aerobic respiration after birth [1]. In mice, lung organogenesis, initiated at E9.5, is subdivided into five stages, including the embryonic stage (E9.5–E12.5), the pseudoglandular stage (E12.5–E16.5), the canalicular stage (E16.5–E17.5), the saccular stage (E17.5–Postenatal (P)5)), and the alveolar stage (P5–P30) [2]. During the pseudoglandular stage, respiratory trees are generated and pulmonary vasculature starts to develop. Terminal bronchioles expand to form respiratory ducts and sacs at the canalicular stage. The following saccular stage is characterized by the thinning of mesenchyme and the differentiation of the distal pulmonary epithelium, which is composed of two types of cells: squamous alveolar epithelial type I cells (AECIs) and cuboidal alveolar epithelial type II cells (AECIIs) [1]. The modeling of saccules into alveoli occurs at the alveolar stage, and during this stage AECIIs undergo marked developmental changes to secrete surfactant proteins [3].

Several critical signaling pathways have been shown to be involved in regulating mouse embryonic lung development, including the Wnt, Shh (Sonic hedgehog), Bmp (bone morphogenetic protein), Notch, and FGF (fibroblast growth factor) signaling pathways. These pathways interact to synergistically regulate embryonic lung development [1,4]. Among them, the FGF signaling pathway plays critical roles in embryonic lung development by regulating the interactions between epithelial and mesenchymal cells [5,6,7,8]. FGFs regulate cellular proliferation, differentiation, migration and survival by binding to FGF receptors (FGFR1, FGFR2, FGFR3, FGFR4) and initiating intracellular signaling pathways, including the well-characterized RaS-MAPK and PI3K/AKT pathways [9,10].

The FGF family comprises at least 22 members, among which, Fgf10—a ligand expressed and released by the adjacent lung mesenchyme—specifically binds and activates Fgfr2b in the epithelium [11,12,13,14]. Fgf10–Fgfr2b signaling has been shown to be essential for lung bud formation and controlling the branching morphogenesis of the lung [4,11,15,16]. The importance of Fgf10 in lung development has been illustrated by the phenotypes of *Fgf10* knockout mice, which die shortly after birth due to the complete absence of lungs [5,8,10,17]. Fgf10 was first detected at around E9.5 when the primary lung buds start to emerge, and its expression level continues to increase during the pseudoglandular stage (E12.5–E16.5), suggesting that Fgf10 plays a key role in branching morphogenesis [18,19,20]. After the late pseudoglandular stage (E16.5), the expression level of Fgf10 gradually declines and is maintained at a certain level to support postnatal alveolar development, lung homeostasis, and lung injury [21,22]. It has been shown that the precise control of Fgf10 expression is important for controlling the balance between cell proliferation, differentiation, and cell lineage determination. Fgf10 hypomorphic mice with significantly reduced Fgf10 expression levels exhibited marked AECIIs defects compared to wild-type mice [23]. However, the overexpression of Fgf10 from E15.5 to E18.5 in transgenic mice has been shown to block alveolar epithelial differentiation [24,25,26]. Altogether, these works of research indicate that Fgf10 expression and signaling are delicately controlled and important for lung development [27]. However, whether Fgf10 expression might abnormally increase due to certain gene aberrant expression and lead to pulmonary hypoplasia in vivo remains unclear.

Mammalian Kub3 was first discovered to be a Ku70-binding protein by yeast-hybrid analysis [28]. The protein sequences of Kub3 are highly conserved in yeast, zebrafish, mice and humans. *KUB3* is mapped to chromosome 12q14.1 in humans, with 6 exons coding 246 amino acids. *KUB3*, also termed *XRCC6BP1* (X-ray repair cross-complementation group 6 binding protein 1), has been found to be amplified and overexpressed in human gliomas [29]. By further analysis of the amplicons on chromosome 12q13-21 containing *KUB3* in several independent cases of recurrent glioblastoma, researchers have revealed that the amplification of *KUB3* is associated with a significantly shorter survival time for glioblastoma patients [29]. The binding of Kub3 to Ku70 suggests the role of this protein in DNA double-strand break (DSB) repair. It has been reported that glioblastoma cell cultures with endogenous *KUB3* gene amplification and elevated Kub3 protein expression show a higher efficiency of double-strand break (DSB) repair after ionizing radiation, indicating the association between *KUB3* amplification and expression with DSB repair efficiency [30]. Although these studies have elucidated the function of KUB3 in relation to DSB repair in glioblastoma cells and patients, the physiological roles of Kub3 in normal mammalian cells remains unknown. In the present study, for the first time, we explore the physiological functions of Kub3 by using *Kub3* gene knockout (KO) mice. *Kub3* KO mice die between E18.5 and before birth or die within 6 h of birth. We further elucidated that *Kub3* deletion results in abnormal lung morphogenesis and pulmonary atelectasis, and that abnormal lung development results from the aberrant cell differentiation of the alveolar epithelial cells and the immaturation of type II alveolar epithelial cells due to disturbed FGF signaling. These findings depict the physiological role of Kub3 in prenatal lung development, advancing our knowledge of regulatory factors in lung development and providing new thought for exploring the mechanisms of disease related to perinatal lung development.

## 2. Results

### 2.1. Kub3 Gene Knockout in Mice Results in Perinatal or Neonatal Lethality

The tissue distribution assay by Western blot showed that Kub3 was expressed in most key organs in adult mice, including the heart, liver, spleen, lung, and kidney, with especially high expression in the lungs (Figure 1A), suggesting that Kub3 might play important physiological roles in mice. To further explore the physiological functions of Kub3, we generated *Kub3* gene knockout (KO) mice lacking exon 2 to exon 5 of the *Kub3* gene through homologous recombination (Figure 1B). Quantitative RT-PCR (qRT-PCR) and Western blot assays showed a loss of Kub3 expression in homozygous KO embryos at E18.5 (Figure 1C,D). In situ hybridization analysis further indicated a total absence of *Kub3* in the KO lungs at E18.5, whereas the control lungs displayed obvious *Kub3* staining in the epithelium and mesenchymal cells (Figure 1E). The offspring of the knockout mice were investigated. Surprisingly, only nine Kub3-deletion homozygous pups (KO) were born, but all died within 6 h after birth (P1(<1 d)) and all other homozygotes died before birth (P(>1d)) (Table 1A), indicating that KO pups died in utero or immediately after birth. To determine at which stage the homozygous KO embryos died, we analyzed the KO embryo ratios at different embryonic stages. After the heterozygotes (HTs) intercrossed, the result showed that the ratio of genotypes (WT: HT: KO = 1:2:1) were in accordance with Mendelian inheritance at E15.5 and E18.5 (Table 1B). These data indicated that *Kub3* KO pups died between E18.5 and before birth or immediately after birth. Taking all the data together, *Kub3* gene knockout in mice resulted in perinatal or neonatal lethality.

### 2.2. Kub3 Deletion Results in Abnormal Lung Morphogenesis

The morphology of embryos at different stages were further investigated. At E12.5, no obvious difference of the shape was observed between WT and KO fetuses. At E15.5 and E18.5, the KO fetuses displayed a smaller size and were a little whiter compared with the WT littermates, but no obvious difference was observed between HT and WT fetuses (Figure 2A). As KO fetuses died after E18.5, the morphological structure of the E18.5 embryos were investigated by hematoxylin and eosin (H&E) staining. The result displayed notable abnormalities in the morphological structure of the KO lungs compared with the WT lungs, showing an increased thickness of the mesenchymal septa and lesser amounts of alveoli (Figure 2B), whereas no obvious change was observed in the heart, liver, spleen, or kidney at E18.5 (Appendix A). Moreover, no obvious change in morphological structure was observed among the lungs in WT, HT or KO embryos at E15.5 (Appendix A). These results suggest that the deletion of *Kub3* might cause abnormal embryonic lung development at E18.5. The morphology of E18.5 embryonic lungs was further investigated. The results showed that the KO lung was smaller and paler than the WT and HT lungs (Figure 2C). qRT-PCR and Western blot analysis indicated that, in E18.5 WT embryos, Kub3 was highly expressed in the lung tissue compared to the other tissues (Figure 2D,E). These results indicate that a deficiency in Kub3 might lead to abnormal embryonic lung development.

### 2.3. Kub3 Deletion Results in Pulmonary Atelectasis in E18.5 Lungs

To further investigate whether the deletion of *Kub3* might cause abnormal embryonic lung development, the physiology status and morphological structure of *Kub3* KO embryonic lungs at E18.5 were further examined. In order to examine the lung physiological status, a floating experiment was applied to the lungs. WT and HT lungs showed normal floating in PBS, but KO lungs sank in PBS buffer (Figure 3A), suggesting that KO lungs were not inflated with air, which might be caused by pulmonary atelectasis. Consistently, histological analysis displayed morphology atelectasis with a collapse of alveolar space, increased thickness of the mesenchymal septa, and lesser amounts of alveoli in the KO lungs, while WT lungs exhibited normal lung structure with expanded alveolar space (Figure 3B), as indicated by the significantly decreased interalveolar distance measured by the mean linear intercept (MLI) (average number of MLI: WT, 51.9 ± 10.1; KO, 15.9 ± 5.9) and the decreased number of distal alveoli estimated by radial alveolar counts (RACs) (average number of RACs: WT, 19.7 ± 4.7; KO, 5.1 ± 2.9) (Figure 3C). Furthermore, the pulmonary atelectasis of the KO lung was not due to the dysfunction of the trachea, as indicated by the normally developed trachea structure of the KO lung (Figure 3D).

As no obvious change in lung morphological structure was observed between KO and WT lungs at E15.5 analyzed by H&E staining embryos (Appendix A), we further investigated at what stage lung defects were first evident by examining the lungs from the pseudoglandular stage to saccular formation (E15.5 to E18.5). During the period of lung development at E15.5, distinct columnar or cuboidal epitheliums were evident in both WT and KO lungs, and no significant difference in the number of distal epithelial buds was seen between KO and WT lungs (Figure 3E,F), which was consistent with the enlarged images of the lungs in the embryos at E15.5. During the late canalicular stage of lung development at E17.5, the WT lung displayed narrower terminal buds, as expected, but the KO lung showed a dense appearance and little saccular expansion (Figure 3G), suggesting that the lung defects of KO were evident from the canalicular stage (E17.5), i.e., late embryonic lung development.

### 2.4. Kub3 Deletion Has No Effect on Lung Cell Proliferation and Cell Apoptosis

To determine the basis of the hypercellularity of mesenchymal septa seen in *Kub3*-KO lungs, we assessed whether this was due to a combination of accelerated proliferation and/or defective apoptosis. PCNA (proliferating cell nuclear antigen) is a molecular marker for cell proliferation. Immunofluorescence staining and WB analysis results showed that the *Kub3* KO lungs displayed similar numbers of PCNA-positive cells in mesenchymal and epithelial cells and similar protein levels of PCNA compared with WT lungs at E18.5 (Figure 4A,B) and at E15.5 (Appendix A), suggesting that *Kub3* deletion has no effect on cell proliferation. In addition, no obvious change in apoptosis was observed between KO and WT lungs at E18.5, as indicated by the rare terminal deoxynucleotidyl transferase dUTP nick end labeling (TUNEL)-positive cells and undetectable cleaved-caspase 3 in the KO lungs as well as in the WT lungs at E18.5 (Figure 4C,D) and at E15.5 (Appendix A). Collectively, these data suggest that the abnormal lung morphology and structure in *Kub3* KO lungs were not due to abnormal lung cell proliferation or apoptosis.

### 2.5. Kub3 Deletion Impairs the Differentiation of Alveolar Epithelial Cells and the Maturation of AECIIs

During the saccular stage (E17.5-P5), progenitor epithelial cells proliferate and differentiate into type I alveolar epithelial cells (AECIs) and type II alveolar epithelial cells (AECIIs). The differentiation and maturation of alveolar epithelium cells were further investigated. Compared with WT controls, E18.5 Kub3 KO lungs displayed decreased AECI differentiation markers, including Aqp5 (aquaporin-5) and Pdpn (podoplanin), and decreased expression levels of AECII differentiation markers, including SP-A, SP-B, SP-C, SP-D, and ATP-binding cassette A3 (Abca3) (Figure 5A,B). Consistent with these observations, immunostaining analysis displayed a dramatically decreased amount of Aqp5 and SP-C in KO lung compared with the WT (Figure 5C). These observations suggest that the differentiation of alveolar epithelial cells is impaired in the absence of Kub3.

During the saccular stage of lung development, AECIIs also turn mature. Immature AECIIs store large amounts of glycogen and convert into phospholipids. Mature AECIIs synthesize surfactant-associated proteins (SPs), including SP-A, SP-B, SP-C, and SP-D [31].

SPs, together with phospholipids, form surfactant complexes that are stored in lamellar bodies (LBs). As shown above, our results showed that the amount of SPs decreased in KO lungs (Figure 5A–C). Further Periodic acid–Schiff (PAS) staining showed an excessive accumulation of glycogen in E18.5 KO lungs (Figure 5D). Further ultrastructural analysis by transmission electron microscopy showed that the cuboidal AECIIs in the KO lung contained abundant large glycogen deposits but fewer LBs (Figure 5E). These results indicate that *Kub3* deletion also leads to the aberration of the maturation of AECIIs. Taking all the data together, these results demonstrate that *Kub3* deficiency impairs the differentiation of alveolar epithelial cells and the maturation of AECIIs.

### 2.6. Kub3 Deletion Results in the Abnormal FGF Signaling Pathway

The development of mouse embryonic lung has been shown to be regulated by several critical signaling pathways, including the Wnt, Shh, Bmp, Notch, and FGF signaling pathways. The aberration of the pathways might result in abnormal lung development [1,4]. To assess the molecular mechanism of abnormal embryonic lung development in *Kub3* KO lungs at E18.5, the mRNA levels of the marker genes of the critical regulatory signaling pathways in E18.5 lung tissues were determined by qRT-PCR. The results showed no obvious difference in the marker genes of the Wnt, Shh, Bmp or Notch signaling pathways between the KO and WT lungs at E18.5 (Figure 6A–D), which was confirmed by the Western blot analysis results (Appendix A). However, compared with WT lungs, the expression of Fgf10 (ligand) and Etv4 (target gene) in the FGF signaling pathway increased in the KO lung at E18.5 (Figure 6E,F). Furthermore, Spry2, an antagonist of the FGF signaling pathway, decreased in the KO lung at E18.5 (Figure 6E,F). These results suggest that Kub3 deficiency leads to abnormal FGF signaling at E18.5. FGF10 binds to FGFR2b and leads to three pathways, including the Ras-MAPK pathway, the PI3K-AKT pathway and the PLCγ pathway [16]. Our results showed that Spry2 and Etv4, marker proteins of the Ras-MAPK pathway, showed changes in the KO lung compared with the WT at E18.5. However, no obvious changes in the target genes of the PI3K-AKT pathway or the PLCγ pathway were observed between the WT and KO lung at E18.5 (Appendix A). Moreover, the immunofluence results of Fgf10, p-Erk, and Etv4, the important proteins in Ras-MAPK, displayed increased expression levels (Figure 6G) which were consistent with the Western blot results. However, no obvious changes in the mRNA levels of Fgf10, Etv4 or Spry2 between WT and KO at E15.5 were observed (Appendix A). These results indicate that Kub3 deficiency results in abnormal FGF results in abnormal FGF signaling through the Ras-MAPK pathway in *Kub3* KO embryonic lungs at E18.5.

To complement the genetic evidence that Kub3 regulates FGF signaling, cell-based assays were employed. The results revealed that the knockdown of KUB3 in the HEK293 cells resulted in increased levels of FGF10, p-ERK and ETV4 (Figure 6H), which was consistent with the results of the *Kub3* KO lung tissues and suggests that Kub3 is critical for FGF signaling transduction in mammalian cells.

## 3. Discussion

Kub3 is a highly conserved protein expressed in all eukaryotic cells, from yeast to plants and human beings, but it is poorly characterized in mammalian cells. Previously, Zeng et al. [32] revealed that its yeast homolog (also called Atp23) was a metalloprotease required for the maturation of subunit 6 precursor and the assembly of yeast mitochondrial ATP synthase. Furthermore, researchers found that Kub3 was overexpressed in human gliomas, and glioblastoma cell culture with elevated Kub3 protein expression showed higher double-strand break (DSB) repair efficiency after ionizing radiation [29,30], indicating that a higher expression of Kub3 is associated with a higher efficiency of DSB repair in human glioma cells. However, the physiological roles of Kub3 in normal mammalian cells remain unknown. In this study, we first revealed a role of Kub3 in mouse late embryonic lung development by using *Kub3* gene knockout mice with the deletion of exons 2–5 from the *Kub3* sequence. The *Kub3* KO mice exhibited abnormal lung morphogenesis and pulmonary atelectasis at E18.5. Kub3 deficiency impaired the differentiation of alveolar epithelial cells and the maturation of type II epithelial cells at E18.5. Further characterization displayed that *Kub3* deficiency causes an abnormal FGF signaling pathway.

Fetal lung development includes the early stage of pulmonary branching morphogenesis (pseudoglandular stage), the late stage of alveolar formation (canalicular stage and saccular stage) and the stage of alveolar remodeling (alveogenesis stage) [33]. Our results showed that Kub3 deficiency caused aberration of the alveolar formation rather than the earlier branching morphogenesis, as evidenced by the comparable development between WT and *Kub3* KO lungs at E12.5 (the early pseudoglandular stage) and E15.5 (the late pseudoglandular stage). However, compared with WT, KO lung displayed a denser appearance and smaller saccular expansion at E17.5 (the late canalicular stage), the collapse of the alveolar space, a thicker mesenchymal space, and smaller amounts of alveoli at E18.5 (the saccular stage). This was further evidenced by the observation of undisturbed cell proliferation and apoptosis in *Kub3* KO lungs at E15.5 and E18.5. During the saccular and alveogenesis stages, epithelial progenitor cells differentiate to AECIs and AECIIs [34]. AECIIs become mature by undergoing ultrastructural and biochemical changes, including the depletion of glycogen, increased numbers of LBs, and the secretion of surfactant into the air spaces [1,35]. Our results displayed the decreased differentiation of AECIs and AECIIs and the immaturation of AECIIs in KO lungs at E18.5, as evidenced by the lesser expression of the marker proteins of the AECIs and AECIIs and increased amounts of glycogen and less LBs in the KO lungs at E18.5. Therefore, these results indicate that Kub3 plays an important role in perinatal pulmonary differentiation and maturation after the late saccular stage.

It is well known that Fgf10 plays an important role in mouse lung development. Fgf10 also plays roles in epithelial–mesenchymal transition, the repair of tissue injury, and embryonic stem cell differentiation [2,16,36]. *Fgf10* knockout mice die shortly after birth due to the complete absence of lungs, as well as forelimbs and hindlimbs [1,2]. It has been demonstrated that Fgf10 hypomorphic embryos (with about 20% of the normal Fgf10 expression) exhibit major defects in different mesenchymal cell types, including ASMCs (airway smooth muscle cells), endothelial cells, and alveolar MYFs (myofibroblasts) [37]. Low expression levels of Tbx4 reduce *Fgf10* transcript levels and impair lung bud formation [38]. Meanwhile, it was also found that the overexpression of Fgf10 in transgenic mice from E15.5 until E18.5 blocked alveolar epithelial differentiation [24,25,26]. However, whether the increased expression of Fgf10 due to certain gene aberrant expression might cause abnormal lung development remains unknown. In our study, we revealed that Kub3 deficiency resulted in the aberration of pulmonary development after E17.5 and the decreased differentiation of AECIs and AECIIs and the immaturation of AECIIs at E18.5. Moreover, we also found increased Fgf10 expression and Ras-MAPK signaling in the *Kub3* KO lung at E18.5. Therefore, our results suggest that increased Fgf10 expression at E18.5 due to *Kub3* deficiency causes the disturbed differentiation of alveolar epithelial cells and the aberration of late embryonic lung development in mice. However, how Kub3 regulates Fgf10 level remains unknown. We tried determining whether Kub3 interacts directly with Fgf10 by co-immunoprecipitation, but the complex of Kub3 and Fgf10 was not detected. It suggested that Kub3 might regulate Fgf10 levels by an unknown intermediate protein, which is worth further study.

In summary, we have uncovered the physiological role of Kub3 in late embryonic lung development, during which Kub3 is indispensable for the differentiation of alveolar epithelial cells and the maturation of AECIIs by regulating the FGF signaling pathway. This advances our knowledge of mouse lung development and provides new thought for exploring the mechanisms of disease related to perinatal lung development.

## 4. Materials and Methods

### 4.1. Animals

The *Kub3* knockout mice used for the experiments were of C57BL/6N background and were originally produced by Cyagen. The animals were maintained on a 12 h light/dark schedule in a specific pathogen-free facility. All animal experiments were approved by the Institutional Animal Care and Use Committee (IACUC) of Huazhong University of Science and Technology (IACUC number: 2730). All experimental procedures were performed according to the relevant guidelines and regulations set by the IACUC.

### 4.2. Generation and Genotyping of the Kub3 Knockout Mice

The *Kub3* knockout mouse model (C57BL/6N) was created by CRISPR/Cas-mediated genome engineering. The mouse *Kub3* gene has six exons; exon 2 to exon 5 were selected as the target sites. Cas9 mRNA and gRNA generated by in vitro transcription were injected into fertilized eggs for knockout mouse production. The founders were genotyped by PCR followed by DNA sequencing analysis. The positive progenies were backcrossed with positive mice to generate the *Kub3* knockout mice.

A PCR-based method was developed for genotyping KO mice. Mouse DNA was purified from the tail of offspring and used for genotyping by PCR using standard methods as described in [39,40]. A two-step PCR was used for genotyping. For the first step, PCR, Primer-F (5′-TCACCCTGGGTTCTCAGCGAGA-3′), and Primer-R (5′-ATCTGCCCAGCCAAAGACTGAACTTT-3′) were used. No bands were seen of wild-type (WT) genotype, and a 950bp band could be seen for the heterozygous or homozygous genotypes. To distinguish the heterozygous and homozygous genotypes, for the second step of the PCR, three primers including Primer-F (5′-TCACCCTGGGTTCTCAGCGAGA-3′), Primer-R (5′-ATCTGCCCAGCCAAAGACTGAACTTT-3′), and Primer-Wt/He-R (5′-CACCCAAGCAACAGGCACCAC-3′) were used. A 590bp band could be seen for WT, which was a positive control. A 950bp band could be seen for the homozygous genotype, and both 590bp and 950 bp bands could be seen for the heterozygous genotype.

### 4.3. Histological Analyses

Lung tissues were fixed in 4% paraformaldehyde overnight, processed with 70% ethanol, and embedded in paraffin. These paraffin embedded tissues were sliced into 5 μm thick sections for hematoxylin and eosin (H&E) and Periodic acid–Schiff (PAS) staining using commercial kits (G1008, Servicebio, Wuhan, China). The TUNEL assay to detect cell apoptosis was performed on paraffin sections using the commercial kits (G1501, Servicebio, Wuhan, China). Pictures were taken on a Nikon microscope with a DS-U3 camera.

### 4.4. In Situ Hybridization

Anti-sense RNA probes labeled with digoxigenin for *Kub3* were generated and samples were processed as previously described [41]. Briefly, fresh embryonic lung tissues were dissected, washed, and fixed in 4% paraformaldehyde (prepared with DEPC water) overnight, paraffin embedded, sectioned at 7 μm, and transferred onto superfrost glass slides. The sections were deparaffinized, rehydrated, fixed, and treated with proteinase K (20 μg/mL) before being hybridized overnight with anti-sense RNA probes (3 μg/mL) at 70 °C. The sections were washed and blocked with anti-Digoxigenin-AP (200-032, Jackson, Bar Harbor, ME, USA) at 37 °C for 40 min. Photos were taken with a positive fluorescence microscope.

### 4.5. Immunofluorescence

Immunofluorescence (IF) was performed manually according to Camolotto et al. [42]. Briefly, lung tissues were fixed overnight in 4% paraformaldehyde and washed in PBS followed by serial dehydration with ethanol prior to embedding. Tissues were sliced into 6 μm thick sections, deparaffinized, rehydrated, and boiled for 22 min with Antigen Unmasking Solution (G1202, Servicebio, Wuhan, China). Slides were subsequently blocked in blocking buffer (3% BSA) for 1 h, incubated with primary antibodies at 4 °C overnight, and washed with PBS. FITC, Alexa-488 or Cy3-conjugated secondary antibodies were used to visualize the protein of interest via fluorescence. The following primary antibodies were used: PCNA (60097-1-Ig, Proteintech, Wuhan, China), SP-C (10774-1-AP, Proteintech, Wuhan, China), AQP5 (sc-514022, santa cruze, Dallas, USA), FGF10 (863308, ZENBIO, Chengdu, China), p-ERK (AF1015, Affinity, Jiangsu, China), ETV4 (10684-1-AP, Proteintech, Wuhan, China).

### 4.6. Transmission Electron Microscopy

Lung tissues were fixed in TEM fixative (G1102, Servicebio, Wuhan, China) and post-fixed in 1% osmium tetroxide in the dark for 2 h at room temperature. The samples were rinsed in phosphate buffer (PB), dehydrated in ethanol, and embedded in resin. Ultrathin sections (60 to 80 nm) were cut on the ultramicrotome. Ultrathin sections were stained with 2% uranium acetate-saturated alcohol solution for 8 min, rinsed in 70% ethanol and ultra-pure water, stained with 2.6% lead citrate by avoiding CO_2_ for 8 min, and then rinsed with ultra-pure water. The samples were examined with an EM208 transmission electron microscope (Philips Electron Optics, Eindhoven, Netherlands).

### 4.7. Hydrostatic Lung Test and Trachea Alcian Blue–Alizarin Staining

Embryonic lung tissue attached to the trachea was taken out, washed in PBS and placed into PBS buffer, and the floatation of the lungs was observed [43]. For the trachea Alcian blue–alizarin staining, lung tissues with trachea were washed with PBS and fixed in 20% ethanol; then, and the samples were rinsed in water, incubated in Alcian blue dye for 15 min, and then washed in running water. The samples were observed under the microscope and images were taken.

### 4.8. Cell Culture and Transfection

HEK293 cells (an embryonic cell line that was isolated in 2008 from the kidney of a human embryo) were generously provided by Prof. Jing Yu Liu (Institute of Neuroscience, Chinese Academy of Sciences, Shanghai, China) and cultured in Dulbecco’s Modified Eagle Medium (DMEM; Gibco, Los Angeles, CA, USA) supplemented with 10% (*v*/*v*) fetal bovine serum (FBS) (Gibco, Los Angeles, CA, USA) and 50 mg/mL penicillin/streptomycin (Gibco, Los Angeles, CA, USA) at 37 °C with 5% (*v*/*v*) CO_2_. The transfection of cells with siRNA was performed by using Lipofectamine2000 (Invitrogen, Los Angeles, CA, USA) according to the manufacturer’s instructions. The sequence of the siRNA specific for *KUB3* was 5′-CCTTAGTGGAGACTGCTCA-3′.

### 4.9. Tissue Total RNA Isolation and Quantitative Real-Time RT-PCR

Total RNA was extracted from mouse embryonic lungs using TRIzol^®^ Reagent (TaKaRa Bio, Tokyo, Japan) according to the manufacturer’s manual. RNA was converted to cDNA using the HiScript II Reverse Transcriptase Kit (Vazyme Biotech, Nanjing, China). Quantitative real-time polymerase chain reaction (qRT-PCR) was performed using SYBR Green Master Mix-High ROX Premixed (Vazyme Biotech, Nanjing, China) according to the manufacturer’s instructions in a Stepone Plus system (BioRad, Los Angeles, CA, USA). Gene expression was calculated using the 2^−ΔΔCt^ method, in which GAPDH was used as an internal reference for each sample. All reactions were performed in triplicate. The primers used for qPCR were shown in Appendix A.

### 4.10. Western-Blot Analysis

Fresh lung tissues and cultured cells were lysed on ice using RIPA buffer (50 mM HEPES, 140 mM NaCl, 1 mM EDTA, 1% Triton X-100, 0.1% sodium deoxycholate and 0.1% SDS) supplemented with protease and phosphatase inhibitor cocktails (Roche, Basle, UK). Protein extracts were clarified and concentration was measured by BCA method. A 40 μg of protein extract was separated by SDS-PAGE and western- blot analysis was performed as previously described [44]. Antibodies used for western-blot analysis include Kub3 (Lab made), SP-A (11850-1-AP, Proteintech, Wuhan, China), SP-B (13664-1-AP, Proteintech, Wuhan, China), SP-C (10774-1-AP, Proteintech, Wuhan, China), SP-D (11839-1-AP, Proteintech, Wuhan, China), AQP5 (sc-514022, santa cruze, Dallas, USA), Fgf10 (863308, ZENBIO, Chengdu, China), Spry2 (11383-1-AP, Proteintech, Wuhan, China), p-ERK (AF1015, Affinity, Jiangsu, China), ERK (AF0155, Affinity, Jiangsu, China), ETV4 (10684-1-AP, Proteintech, Wuhan, China), PCNA (60097-1-Ig, Proteintech, Wuhan, China), Caspase-3 (19677-1-AP, Proteintech, Wuhan, China), BMP4 (UM500038, OriGene, Wuxi, China), Gli1 (66905-1-AP, Proteintech, Wuhan, China), Hes1 (ab108937, Abcam, Cambridge, UK), PI3K (20584-1-AP, Proteintech, Wuhan, China), p-AKT (#4060, Cell Signaling, Boston, MA, USA), t-AKT (#4691, Cell Signaling, Boston, MA, USA), and GAPDH (13664-1-AP, Proteintech, Wuhan, China). For signal development, Supersignal West Femto Substrate kit (Thermo Fisher Scientific, Boston, MA, USA) was used, followed by image acquisition using X-ray film or Protein Simple Fluor Chemical imaging system (BioRad, Los Angeles, CA, USA). Image J was used for band intensity quantitation.

## 5. Conclusions

This study uncovered an essential physiological role of Kub3 in prenatal lung development by generating and characterizing *Kub3* gene knockout mice. We revealed that *Kub3* gene knockout mice die between E18.5 and before birth, or die within 6 h after birth. Kub3 deletion results in abnormal lung morphogenesis and pulmonary atelectasis at E18.5, and abnormal lung development results from the aberrant differentiation of alveolar epithelial cells and the immaturation of type II epithelial cells due to a disturbed FGF signaling pathway. This advances our knowledge of the regulatory factors in embryonic lung development and provides new thought for exploring the mechanisms of disease related to perinatal lung development.

## Figures and Tables

**Figure 1 ijms-23-06014-f001:**
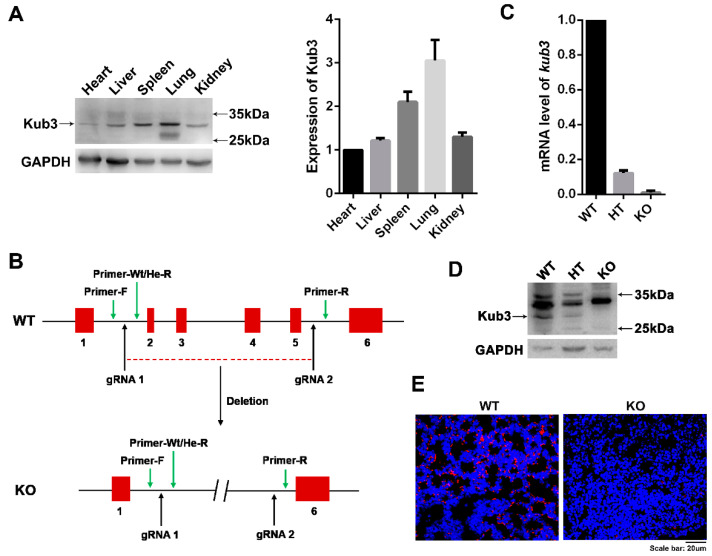
*Kub3* gene knockout in mice results in perinatal or neonatal lethality. (**A**) Tissue distribution of Kub3 expression in adult mice examined by Western blot (WB). Total protein lysates were isolated from different organs of adult mice. Western blot was performed with antibodies against Kub3. GAPDH is the internal control. The ratios of Kub3 to GAPDH expression are represented by the graphs (*n* = 3). (**B**) Schematic diagram depicting the deletion of exons 2-5 from the wild-type *Kub3* gene and creating *Kub3* knockout mice. Red squares represent the exons of *Kub3* and blue arrows indicate the sites of the primers for genotype identification. (**C**) Quantitative RT-PCR analysis of *Kub3* mRNA levels in E18.5 embryos. GAPDH was used as an internal control for normalization (*n* = 3). (**D**) WB analysis of Kub3 protein level in E18.5 embryos. Total protein lysates were isolated from embryos of WT, HT and KO at E18.5. GAPDH was used as a loading control. (**E**) In situ hybridization (ISH) analysis of *Kub3* in WT and KO lungs at E18.5. *Kub3*-positive signals are shown as red fluorescence in cells. Nuclei were stained blue with 4′,6-diamidino-2-phenylindole (DAPI). Scale bar: 20 µm.

**Figure 2 ijms-23-06014-f002:**
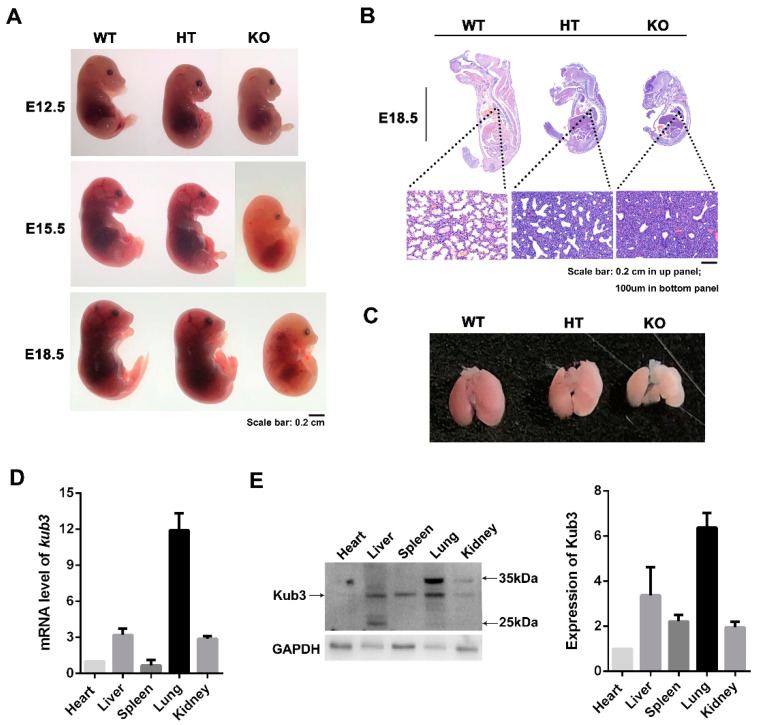
*Kub3* deletion results in abnormal lung morphogenesis. (**A**) Gross morphology of embryos at different stages. WT, HT and KO embryos of different development stages (E12.5, E15.5, E18.5) were dissected and photos were taken. Scale bar: 0.2 cm. (**B**) Morphological structure of E18.5 embryos examined by hematoxylin and eosin (H&E) staining. Images of lung tissues are enlarged in the bottom panel. Scale bar: 0.1 cm in upper panel; 200 µm in bottom panel. (**C**) Gross morphology of embryonic lungs at E18.5. Lungs were dissected from E18.5 embryos and photos were taken. Scale bar: 0.2 cm. (**D**) qRT-PCR analysis of the mRNA levels of *Kub3* in different tissues of WT embryos at E18.5. GAPDH is the internal control (*n* = 3). (**E**) Kub3 protein level in different tissues in WT E18.5 embryos by Western blot analysis. Total protein lysates were isolated from different tissues of WT E18.5 embryos. Western blot was performed with an antibody against Kub3. GAPDH is the internal control. The ratios to GAPDH expression are represented by the graph (*n* = 3).

**Figure 3 ijms-23-06014-f003:**
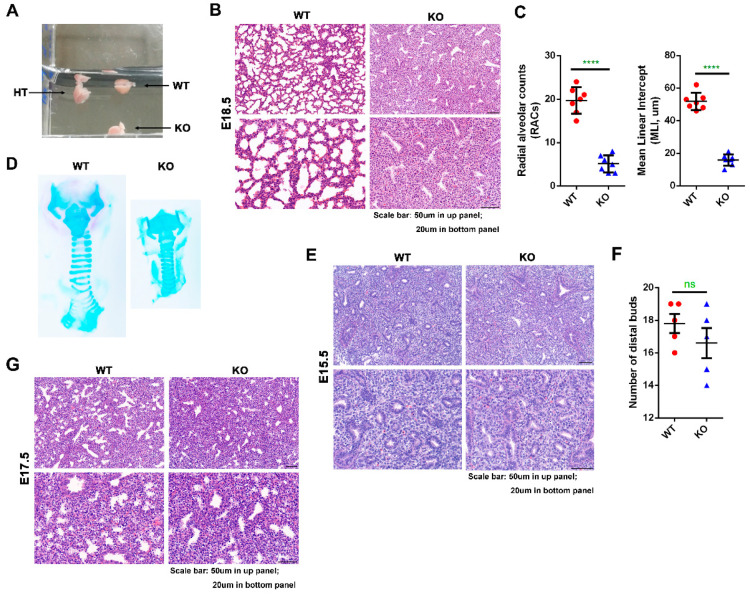
*Kub3* deletion results in pulmonary atelectasis during embryonic development. (**A**) Floating assay for E18.5 lungs. WT, HT and KO lungs were dissected and put in PBS buffer. WT and HT lungs floated in PBS buffer, but *Kub3* KO lungs sank due to a lack of air inflation. (**B**) Representative images of H&E-stained lung sections of WT and KO lungs at E18.5. Scale bar: 50 µm in the upper panel; 20 µm in the bottom panel. (**C**) Morphometric analysis of lung interalveolar distance measured by mean linear intercept (MLI) and number of distal alveolar estimated by radial alveolar counts (RACs) in WT and KO lungs at E18.5 (*n* = 7 per group; **** *p* < 0.0001). (**D**) Alcian blue–Alizarin staining of trachea in the E18.5 lung. The cartilaginous trachea is stained dark blue by Alcian blue. (**E**) Representative images of H&E-stained lung sections of WT and KO lungs at E15.5. Scale bar: 50 µm in the upper panel; 20 µm in the bottom panel. (**F**) Morphometric analysis of the number of distal epithelial buds in E15.5 lungs (*n* = 6 per group. *p* > 0.05, ns: non-significant). (**G**) Representative images of H&E-stained lung sections of WT and KO lungs at E17.5. Scale bar: 50 µm in upper panel; 20 µm in bottom panel.

**Figure 4 ijms-23-06014-f004:**
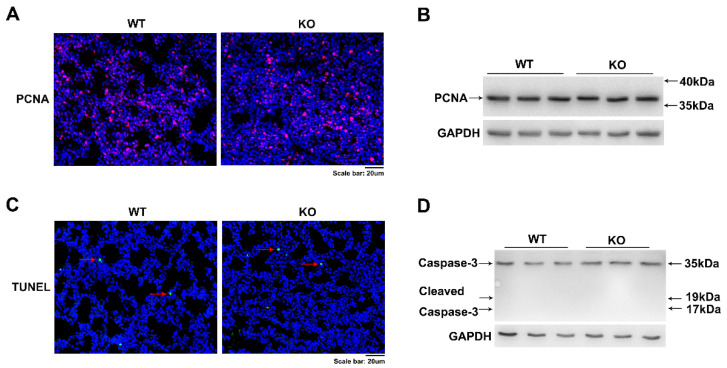
*Kub3* deletion has no effect on lung cell proliferation and cell apoptosis. (**A**) Immunostaining of PCNA in lung sections of E18.5 lungs. Nuclei were stained blue with 4′,6-diamidino-2-phenylindole (DAPI). Scale bar: 20 µm. (**B**) Western blot analysis of PCNA protein level in E18.5 lungs. GAPDH was used as a loading control. (**C**) Cell apoptosis analysis of cells in E18.5 lungs examined by terminal deoxynucleotidyl transferase dUTP nick end labeling (TUNEL) assay. Arrows indicate TUNEL-positive nuclei. Scale bar: 20 µm. (**D**) Western blot analysis of protein levels of caspase-3 and cleaved caspase-3 in the KO lung at E18.5. GAPDH expression is the protein loading control.

**Figure 5 ijms-23-06014-f005:**
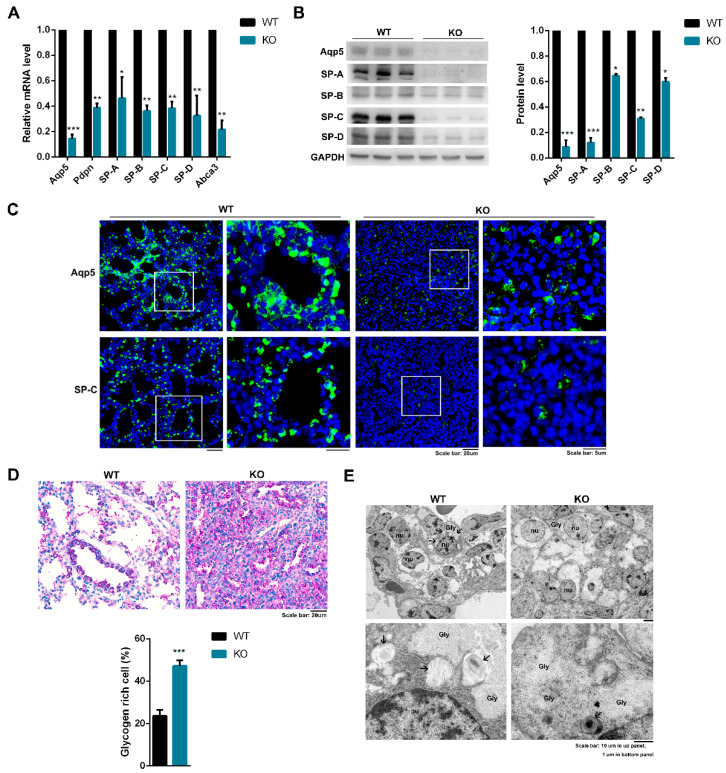
*Kub3* deletion impairs the differentiation of alveolar epithelial cells and the maturation of AECIIs. (**A**) Quantitative RT-PCR analysis of the mRNA levels of marker genes for AECIs and AECIIs in E18.5 lungs. The data are derived from three independent experiments performed in triplicate and are normalized to the GAPDH (*n* = 3. *** *p* < 0.001, ** *p* < 0.01, * *p* < 0.05). (**B**) Immunoblot analysis of protein levels of AECI and AECII markers. GAPDH expression was used as a loading control. The ratios to GAPDH expression are represented by bar graphs (*n* = 3. *** *p* < 0.001, ** *p* < 0.01, * *p* < 0.05). (**C**) Immunostaining of Aqp5 and SP-C in lung sections between WT and KO at E18.5. The images on the right enlarged from box regions are in high magnification. Scale bar: 20 µm and 5 µm. (**D**) Periodic acid–Schiff (PAS) staining in the lung sections of E18.5 lungs. Scale bar: 20 µm. The statistical analysis of glycogen-rich cells among alveolar epithelial cells is presented by the bar graph; *** *p* < 0.001. (**E**) Transmission electron microscopic images of E18.5 lungs. Wild-type AECIIs contained numerous lamellar bodies (LBs) (arrows), but lamellar bodies were barely observed in KO lungs, and the cytoplasm was occupied by glycogen. Gly—glycogen deposit; Nu—nucleus. Scale bar: 10 µm in upper panel; 1 µm in bottom panel.

**Figure 6 ijms-23-06014-f006:**
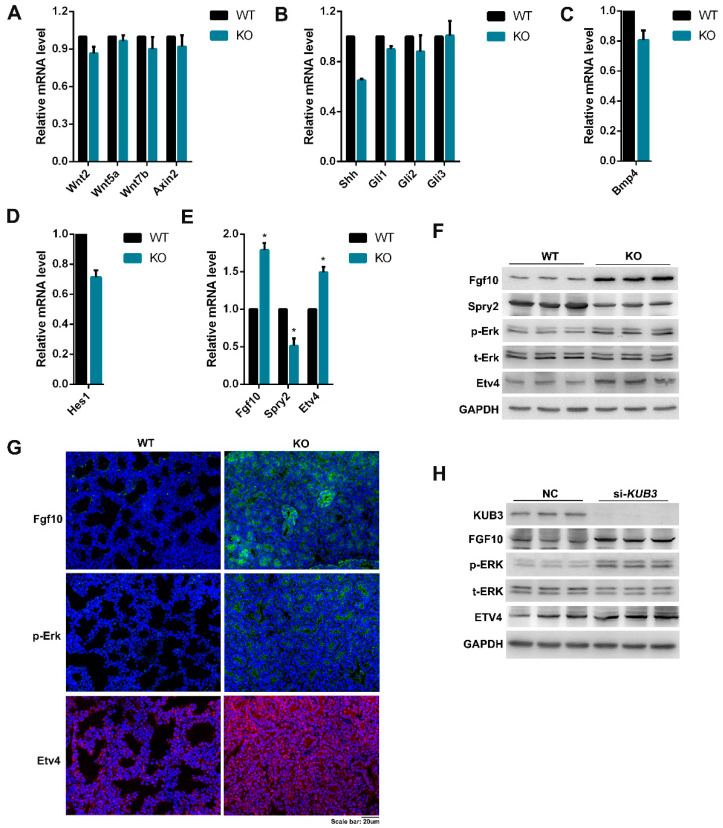
*Kub3* deletion results in an abnormal FGF signaling pathway. (**A**–**E**) Quantitative RT-PCR analysis of the mRNA levels of marker genes for the Wnt, Shh, Bmp, Notch, and FGF signaling pathways, respectively. The data are derived from three independent experiments and are normalized to the GAPDH (*n* = 3. * *p* < 0.05). (**F**) Protein levels of marker proteins for the FGF signaling pathway in E18.5 lungs. Protein lysates were subjected to Western blot performed with the indicated antibodies. GAPDH was used as a loading control. (**G**) Immunostaining of Fgf10, p-Erk, and Etv4 in E18.5 lung sections. (**H**) Western blot analysis of marker proteins for the FGF signaling pathway in HEK293 cells. GAPDH was used as a loading control.

**Table 1 ijms-23-06014-t001:** Genotypes and viability of *Kub3* KO mice (HT × HT).

**(A) Postnatal Offspring of HT Intercrosses**					
		**WT**	**HT**	**KO**	**Total**
P(>1 d)	Actual	28	29	0	57
	Expected	28	56	28	112
P1(<1 d)	Actual	8	14	9	31
	Expected	8	16	8	32
**(B) Embryos of HT Intercrosses**					
		**WT**	**HT**	**KO**	**Total**
E15.5	Actual	11	25	8	44
	Expected	11	22	11	44
E18.5	Actual	11	21	13	45
	Expected	11	22	11	44

E refers to embryonic day; P refers to postnatal day.

## Data Availability

The data presented in this study are available in the article.

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
