# Peer review of "Kub3 Deficiency Causes Aberrant Late Embryonic Lung Development in Mice by the FGF Signaling Pathway"

_ijms, 2022, doi:10.3390/ijms23116014_

Round 1

Reviewer 1 Report

The authors report a novel finding in this manuscript that is unexpected.  They have determined that Kub3 when inactivated in mice alters lung development. Specifically, in the absence of kub3 the differentiation/maturation of AECII fails to occur. Along with this the authors show that Fgf10 levels are increased, which could explain the observed defect. 

The following are my suggestions to improve the quality of this manuscript:

  • Figure 1: please show whole western blot with molecular weight ladder in panel A and D.
  • Figure 1E: it would be good to stain for lung c ell type markers to determine in which cell types Kub3 is normally expressed in. Is it expressed in progenitor cells that differentiate into AECIIs?
  • Figure 2E and Figure 4B, please show the whole western blot along with the molecular weight ladder.
  • The Aqp5 shown in Figure 5C does not appear to stain the cell membrane. It would be good if the authors show higher magnification images.
  • Assessment of additional markers of the progenitor cells that differentiate into AECIIs is needed to conclude that the progenitors fail to differentiate.
  • The expression level of shh and hes1 appear to be lower in the Kub3 mutants (Figure 6). The authors should discuss how changes in the expression of these genes fits into the model of Kub3 functioning mainly by regulating fgf10 levels.
  • It would be interesting for the reader if the authors speculate in the discussion as to the mechanism by which they think Kub3 regulates fgf10 levels.

Reviewer 2 Report

Summary:

The authors generated Kub3 knockout mice to investigate the physiological roles of Kub3 in normal mammalian cells and mice prenatal lung development. They found that that Kub3 knockout mice died as embryos after E18.5 or as newborns immediately after birth. The authors suggest a new thought for exploring the mechanism of disease related to perinatal lung development.

General concerns:

  1. Page 3, lines 108-110: Surprisingly, rarely Kub3 deletion homozygous pups (KO) were born, and all of the homozygotes died before birth [P(>1d)] or die within 6 hours after birth [P1(<1 d)] (Table 1A), indicating that KO pups died in utero or immediately after birth. According to Table 1A, 9 mice were alive within 6 hours after birth [P1(<1 d)]. Please confirm it.
  2. Figure 1C, 1D, and 1E: Please indicate the tissue source.
  3. Figure 3C: RAC and MLI values revealed opposite trend in lung development. Please describe the RAC and MLI values revealed similar trend. Is it due to lungs not well-expanded?
  4. FGF10 is a key positive regulator of lung branching morphogenesis. FGF10 upregulated the expression of mSprouty2 and BMP4 in embryonic lung explant culture (Mailleux et al. Evidence that SPROUTY2 functions as an inhibitor of mouse embryonic lung growth and morphogenesis. Mech Dev 2001;102:81-94). In Figure 6F and FigureS3, Kub3KO mice increased FGF10 expression and decreased Sprt2 and BMP4 expression. Please clarify the disassociation.
